Does elevated atmospheric CO2affect soil carbon burial and soil weathering in a forest ecosystem?

Gonzalez-Meler Miquel A. mmeler@uic.edu 1
Poghosyan Armen 1 2
Sanchez-de Leon Yaniria 1 3
Dias de Olivera Eduardo 1
Norby Richard J. 4
Sturchio Neil C. 1 5
1 Department of Biological Sciences and Department of Earth and Environmental Sciences, University of Illinois at Chicago , Chicago , IL , USA
2 Space Center, Skolkovo Institute of Science and Technology , Moscow , Russia
3 Department of Agro-environmental Sciences, Universidad de Puerto Rico at Mayaguez , Mayaguez , Puerto Rico
4 Environmental Science Division and Climate Change Science Institute, Oak Ridge National Laboratory , Oak Ridge , TN , USA
5 Department of Earth and Environmental Sciences, University of Delaware , Newark , DE , USA
Giordani Paolo
Electronic publication date: 2018 Jul 27
Publication date: 2018
Volume: 6
Electronic Location ID: e5356
Received 2018 Mar 6; Accepted 2018 Jul 11
Copyright: ©2018 Gonzalez-Meler et al.
Copyright year: 2018
Copyright holder: Gonzalez-Meler et al.
License: This is an open access article distributed under the terms of the Creative Commons Attribution License, which permits unrestricted use, distribution, reproduction and adaptation in any medium and for any purpose provided that it is properly attributed. For attribution, the original author(s), title, publication source (PeerJ) and either DOI or URL of the article must be cited.
License URL: https://creativecommons.org/licenses/by/4.0/

Keywords: Soil C, Elevated CO2, Isotope, Temperate forest, Bioturbation, cesium-137, lead-210

Funding: United States National Science Foundation DEB-0919276 NSF-ADVANCE postdoctoral fellowship United States Department of Energy, Office of Science, Biological and Environmental Research program United States Department of Energy DE-AC05-00OR22725 This work was supported by the United States National Science Foundation grant DEB-0919276 to Miquel Angel Gonzalez-Meler and Yaniria Sánchez-de León. Dr. Sánchez-de León was also supported by a NSF-ADVANCE postdoctoral fellowship. The Oak Ridge National Laboratory FACE site was supported by the United States Department of Energy, Office of Science, Biological and Environmental Research program. Oak Ridge National Laboratory is managed by University of Tennessee-Battelle, LLC for the United States Department of Energy under contract DE-AC05-00OR22725. The funders had no role in study design, data collection and analysis, decision to publish, or preparation of the manuscript.

==============================
Most experimental studies measuring the effects of climate change on terrestrial C cycling have focused on processes that occur at relatively short time scales (up to a few years). However, climate-soil C interactions are influenced over much longer time scales by bioturbation and soil weathering affecting soil fertility, ecosystem productivity, and C storage. Elevated CO2can increase belowground C inputs and stimulate soil biota, potentially affecting bioturbation, and can decrease soil pH which could accelerate soil weathering rates. To determine whether we could resolve any changes in bioturbation or C storage, we investigated soil profiles collected from ambient and elevated-CO2plots at the Free-Air Carbon-Dioxide Enrichment (FACE) forest site at Oak Ridge National Laboratory after 11 years of 13C-depleted CO2 release. Profiles of organic carbon concentration, δ13C values, and activities of 137Cs, 210Pb, and 226Ra were measured to ∼30 cm depth in replicated soil cores to evaluate the effects of elevated CO2 on these parameters. Bioturbation models based on fitting advection-diffusion equations to 137Cs and 210Pb profiles showed that ambient and elevated-CO2 plots had indistinguishable ranges of apparent biodiffusion constants, advection rates, and soil mixing times, although apparent biodiffusion constants and advection rates were larger for 137Cs than for 210Pb as is generally observed in soils. Temporal changes in profiles of δ13C values of soil organic carbon (SOC) suggest that addition of new SOC at depth was occurring at a faster rate than that implied by the net advection term of the bioturbation model. Ratios of (210Pb/226Ra) may indicate apparent soil mixing cells that are consistent with biological mechanisms, possibly earthworms and root proliferation, driving C addition and the mixing of soil between ∼4 cm and ∼18 cm depth. Burial of SOC by soil mixing processes could substantially increase the net long-term storage of soil C and should be incorporated in soil-atmosphere interaction models.

Introduction

Soils contain most of the organic carbon in Earth’s “critical zone”, thus formation, transport and degradation of soil organic carbon (SOC) are key factors in the global carbon cycle (Hopkins et al., 2013). Fixation of atmospheric CO2 by plant photosynthesis and the consequent decomposition and release of this organic carbon as CO2 by soil biota are principal factors in the evolution of the SOC pool and the atmospheric concentration of CO2. Soil organic carbon decomposition depends on vegetation, microbial community, molecular composition of the organic matter, mineralogy, moisture, and temperature (Jastrow, 1996; Jastrow, Amonette & Bailey, 2007; O’Brien et al., 2010; Cheng et al., 2014). Climate change forcing factors can directly and indirectly affect soil C stocks, altering the resilience of vegetation and human society to climate change (Jastrow et al., 2005; Hungate & Hampton, 2012; Gonzalez-Meler, Rucks & Aubanell, 2014; Marshall & Gonzalez-Meler, 2016). However, the long-term fate of terrestrial soil C stocks under climate change scenarios may also be a function of soil C transport and burial processes (Chaopricha & Marin-Spiotta, 2014). Transport of SOC within the soil C matrix is difficult to measure but SOC burial has been recognized in playing a role in the responses of the soil C pool to climatic factors ( Lehmann & Kleber, 2015).

Mechanical mixing of soil by bioturbation (the mixing of soil particles by biological agents) can modulate the rate of SOC decomposition by vertical transport, potentially bringing SOC from the surface to depth, and vice-versa (Gabet, Reichman & Seabloom, 2003; Wilkinson, Richards & Humphreys, 2009). This process operates slowly and affects the SOC cycle on centurial time-scales, yet its effects must be taken into account when modeling carbon fluxes at regional or global scales (Koven et al., 2009; Drewniak & Gonzalez-Meler, 2017). Our need to understand the climate feedbacks caused by the alteration of the global carbon cycle is becoming more urgent because of the dramatic increase in atmospheric CO2 caused by anthropogenic activities. Long-term predictions of Earth system responses to global climate change or CO2 increase require a better understanding of soil C processes that operate at multi-decadal time scales (e.g., O’Brien et al., 2010; O’Brien et al., 2013; O’Brien et al., 2015) to model future biosphere feedbacks on atmospheric greenhouse gas composition. Specific mechanistic information on bioturbation in temperate forested ecosystems is limited (Fujiyoshi & Sawamura, 2004; Kaste, Heimsath & Bostick, 2007; Kaste et al., 2011), and available studies generally do not explicitly link long term soil C movement to climate change forcing factors.

Soil biota can alter soil chemical and physical properties in response to climate change and perhaps accelerate soil mixing and C burial rates (Wilkinson, Richards & Humphreys, 2009; Sánchez-de León et al., 2014; Chaopricha & Marin-Spiotta, 2014). Increased soil biological activity in ecosystems exposed to elevated CO2 often increases soil CO2 concentrations ( Gonzalez-Meler & Taneva, 2011), that may cause soil acidification and increased weathering rates (Andrews & Schlesinger, 2001; Bernhardt et al., 2006). Lowering pH and increased plant nutrient uptake may result in loss of soil fertility, affecting the way plants further respond to elevated CO2. Evidence for net loss of metal and cations via leaching has been shown in some elevated CO2 studies (Cheng et al., 2010) but not in others (Oh et al., 2007; Kaste et al., 2011; Duval et al., 2013). The bulk of C and nutrients in the soil is associated with particles, yet it is not well understood how soil particle mixing would determine long-term C storage in a high-CO2 world.

The Free-Air Carbon Dioxide Enrichment (FACE) enrichment experiment at Oak Ridge National Laboratory (ORNL) in eastern Tennessee, USA, provided an opportunity to examine the effects of elevated atmospheric CO2 on SOC and bioturbation in a closed-canopy deciduous forest ecosystem (Norby et al., 1999; Norby et al., 2001). This site has been shown to accrue more soil C at the elevated CO2 conditions when compared to ambient conditions (Jastrow et al., 2005). In addition, elevated CO2 has enhanced root proliferation (Matamala et al., 2003; Iversen et al., 2011; Lynch et al., 2013) and earthworm activity (Sánchez-de León et al., 2014; Sánchez-de León et al., 2018), two major drivers of bioturbation in temperate forest soils (Wilkinson, Richards & Humphreys, 2009).

In conjunction with the soil C cycle and earthworm studies, we measured soil profiles of fallout 137Cs and 210Pb activities, along with those of 40K and 226Ra. Large pulses of 137Cs were introduced into the stratosphere during thermonuclear weapons tests of the 1950s and 1960s, with a well-defined maximum deposition at Earth’s surface occurring in 1963. This surface deposition of 137Cs and other weapons fallout radionuclides provides a globally distributed time horizon in soils and sediments, which has been used widely to determine sedimentation rates and sediment mixing by organisms in soils, lakes and oceans (Guinasso & Schink, 1975; Olsen et al., 1981; Robbins, 1986; Kaste, Heimsath & Bostick, 2007; Kaste et al., 2011). In contrast to the bomb-pulse input of 137Cs, 210Pb is continuously deposited from the atmosphere and is also produced by decay of 226Ra in soil via 222Rn. Because 137Cs and 210Pb are strongly adsorbed to soil particles and are not biologically transformed, they are especially useful as tracers of soil mixing and bioturbation at different shallow soil depths (Bruckmann & Wolters, 1994; Bunzl, 2002a; Bunzl, 2002b; Schuller et al., 2004; Kaste, Heimsath & Bostick, 2007; Kaste et al., 2011). In this study, we apply advection-diffusion models to estimate bioturbation rates from 137Cs and 210Pb profiles in soils of the ORNL FACE site, and use these results along with 226Ra and 40K profiles to compare bioturbation, redistribution of SOC, and potential weathering effects under ambient and elevated-CO2 conditions.

Material and Methods

The CO2 treatment at the ORNL FACE experiment was initiated in 1998 and continued for 12 growing seasons through 2009. The site is contained in a 1.7-hectare sweetgum (Liquidambar styraciflua L.) plantation on the Oak Ridge National Environmental Research Park that was planted with 1-year-old trees in 1988 on an upland terrace of the Clinch River. The FACE experiment comprised five 25-m diameter plots (two elevated CO2 and three control plots), each plot representing a replicate. The CO2 concentration in the elevated CO2 plots was maintained about 150 ppm above ambient during the experiment, at first continuously until 2001, and then only during daylight hours through the end of the experiment in 2009. Soil at the ORNL FACE site is classified as an Aquic Hapludult (Ultisol) that developed from alluvium derived from dolomite, sandstone, and shale. It is a moderately well drained, slightly acidic, silty clay loam soil with high base saturation (Van Miegroet, Norby & Tschaplinski, 1994). Results of the ORNL FACE experiment have been highlighted in several articles (Matamala et al., 2003; Norby et al., 2010; Iversen et al., 2012).

Sampling and sample preparation

Soil samples were collected from the ORNL FACE site ten years into the experiment in September 2008. We used a sharpened steel pipe (4.8 cm diameter) driven into the soil with a nylon-face mallet (Sánchez-de León et al., 2018) to obtain four cores from each of the ambient (control) plots and four from each of the elevated CO2 plots. Intact soil cores were stored frozen and sectioned with a thin ice-core saw (while frozen and the blades cleaned between cuts) as follows: the top 8 cm of each core was sectioned into 1-cm depth increments, and from 8 to ∼20 cm depth the core was sectioned into 2-cm depth increments. Additional soil core samples from 20 to 25 and 25 to 30 cm were collected adjacent to each sampling spot to help constrain the maximum depth of measurable 137Cs activity. No samples were collected below 30 cm depth for these experiments. We compared our samples with soils samples collected in 1997 (prior to the initiation of FACE experiment) and archived. Pre-treatment core samples were for depth ranges of 0–5, 0–15, 15–30, and 30–45 cm from both the ambient and elevated-CO2 plots.

After sectioning the soil cores, rocks and roots were manually removed from each section. Samples were dried at 80 °C, gently crushed and sieved to pass through a 2-mm sieve. Dry bulk densities were calculated from separate samples by comparing the 2 mm-sieved soil dry weight with the core section volume after correction for the occasional small rocks being removed.

Soil organic carbon concentration and stable C isotope ratios

Soil samples were ground to a fine powder for analysis of organic C concentration and stable C isotope ratios. Carbonates were removed before analyses as explained elsewhere (O’Brien et al., 2015). Analyses were performed at the Ecology Stable Isotope Laboratory (UIC) using a Costech ECS 4010 elemental analyzer with a zero-blank autosampler interfaced with a ThermoFinnigan Delta-Plus XL isotope ratio mass spectrometer in continuous flow. Soil organic C concentrations are reported in %C (dry weight basis). The 13C/12C isotope ratios are reported in the conventional delta notation, in units of per mil relative to the standard reference material VPDB (Coplen, 1996), according to: (1) δ13C,‰=R sample∕RV PDB−1×1,000

where R is the atom ratio 13C/12C. Reproducibility of δ13C values is better than ±0.1‰  when compared to international standards.

Gamma spectrometry

Gamma spectrometry was performed at the Environmental Isotope Geochemistry Laboratory (UIC) by using a Canberra model GR3020 reverse-electrode intrinsic Ge detector system interfaced with a DSA-2000 digital spectrum analyzer. Dry homogenized sediment samples (5–10 g) were weighed into aluminum counting cans and these were sealed with Al foil. Gamma activities for 40K, 137Cs, 210Pb, and 226Ra were measured at 1,460.5, 661.6, 46.5, and 186.2 keV, respectively, with cans centered on top of the detector. Detector efficiency was calibrated versus sample weight in the same geometry using the certified standards CANMET DL-1a (U-Th ore diluted in quartz sand) and NIST SRM-4357 (Ocean Sediment). Relative uncertainties of measured activities were less than ±10% for activities >4 Bq kg−1, as calculated from counting statistics incorporating background subtraction and propagated errors. Activities were measured per sample dry mass and normalized to dry bulk density measurements for reporting in units of Bq cm−3 or Bq kg−1.

Bioturbation model based on 137Cs profiles

Mathematical models combining advection and diffusion have been developed to explain the downward movement of 137Cs and the diffusion-like broadening of its profile in soils and sediments (Guinasso & Schink, 1975; Olsen et al., 1981; Robbins, 1986). The steady-state bioturbation model (Eq. (2) in Robbins, 1986) describes the total concentration [C(x, t)] of particle-bound radionuclides in the soil as a function of the vertical distance (x), and time (t). Biological agents and advection explain the downward transport of 137Cs (see Eq. (2)). The biodiffusion coefficient (Db) describes diffusive mixing of bulk soil by biological agents. Transport of 137Cs can also be caused by advective processes involving motion of particles and pore fluid (ν). The net loss or gain of the 137Cs within the soil profile is accounted for by the radioactive decay constant (λ) and the first-order feeding rate constant that describes net transport of 137Cs by moving organisms (γ): (2) ∂C∂t=∂∂xDb∂C∂x−∂∂xνC−λ+γC.

Best-fits for the average ambient and elevated-CO2 137Cs profiles used the following fixed values for the model (Eq. (2)): 0.75 Bq cm−2 for the initial activity of 137Cs (C0) (Hardy et al., 1968; CDC-NCI, 2005); 45 years for time elapsed (t) between 137Cs tracer deposition in 1963 and sample collection in 2008 (or 34 years for 1997); and, 0.023 yr−1 for the 137Cs decay constant (λ). For a pulse-like input of the tracer, the model represented in Eq. (2) has a well-known solution, which has been widely used to describe 137Cs profiles in soils (Van Genuchten & Cleary, 1979; Ivanov et al., 1997; Bossew & Kirchner, 2004; Schuller et al., 2004). (3) Cx,t=C0e−λ+γt1πDbte−x−νt2∕4Dbt−ν2Dbeνx∕Dberfcx+νt2Dbt.

We calculated error-weighted least-squares best fits of Eq. (3) to the data for each of our measured 137Cs profiles by using MATLAB.

Bioturbation model based on unsupported 210Pb profiles

Mathematical models combining advection and diffusion terms to account for downward transport and dispersion of unsupported 210Pb differ from those for 137Cs, because 137Cs is deposited in a pulse-like manner whereas 210Pb is deposited continuously as it is produced from decay of atmospheric 222Rn (Robbins, 1978). We used the following steady-state equation to describe advective-diffusive transport of 210Pb (Kaste et al., 2011), where AZ is the initial activity of unsupported 210Pb at the surface (Bq/kg); AZ is the activity of unsupported 210Pb at depth z (cm); v is the advection rate (4) Az=A0expν−ν2+4λD2Dz.

(cm yr−1); D is the diffusion constant (cm yr−1); and λ is the decay constant of 210Pb (0.031 yr−1).

Results

Soil organic carbon and δ13C profiles

Soil organic carbon content was highest at the surface and decreased with depth (Fig. 1A). The top 6 cm of the elevated-CO2 profiles, on average, have δ13C values significantly lower than in the ambient profiles (Fig. 1B), as indication of SOC inputs since the initiation of the experiment in 1998. As a result of new inputs, the top 2 cm of the elevated-CO2 profiles, on average, have significantly higher SOC content than in the ambient profiles (p < 0.05), but below 2–4 cm depth the average profiles are not significantly different (Fig. 1A). When the δ13C values are compared with the inverse SOC content, it is apparent that the average elevated-CO2 profile is depleted in 13C. The SOC being deposited at the soil surface has a δ13C value of about −38‰  in the elevated-CO2 plot compared with −28‰  in the ambient plot (Fig. 2).

Figure 1 Soil organic carbon vs depth.

(A) depth (cm) vs. soil organic carbon (wt. %) for average core samples from ambient (open circles) and elevated-CO2 plots (filled circles); (B) depth (cm) vs. δ13C (‰) for average core samples from ambient (open circles) and elevated-CO2 plots (filled circles).

Figure 2 δ13C vs. inverse concentration of organic carbon.

Diagram showing δ13C (‰) vs. inverse concentration of organic carbon (1/wt. %) for averages of core profiles from the ambient CO2 plot (open circles) and the elevated-CO2 plot (filled circles). Black lines are 2nd-order polynomial best fits. Shift of the elevated-CO2 profile toward the Y-axis indicates enrichment in organic carbon relative to the ambient CO2 profile. Y-intercepts represent contrasting δ13C values of organic carbon being added to the surface under ambient and elevated-CO2 conditions.

Soil bulk density was lower at the top 5 cm of the soil profile than at the rest of the soil depths (Table 1). Bulk density increased from values of about 0.5 g cm−3 at shallow depths to values greater than 1 g cm−3 at 5–6 cm depth and deeper. This may reflect degradation and mineralization of the litter layer which occurs during the first decades following deposition and produces denser residual material (Kaste et al., 2011).

Table 1 Soil bulk density (kg cm−3) across the soil profile for sections of soil cores collected at ambient and Elevated CO2 plots at the ORNL FACE experiment in 2008.

Values are averages of three and two replicates for ambient and elevated plots, respectively, with standard errors.

Soil depth (cm)	Ambient CO2	Elevated CO2	
0–1	0.58 ± 0.07	0.46 ± 0.07	
1–2	0.83 ± 0.11	0.85 ± 0.01	
2–3	0.97 ± 0.21	1.08 ± 0.31	
3–4	0.96 ± 0.32	1.23 ± 0.25	
4–5	1.16 ± 0.09	1.02 ± 0.01	
5–6	1.27 ± 0.06	1.43 ± 0.13	
6–7	1.43 ± 0.20	1.00 ± 0.19	
7–8	1.26 ± 0.11	1.26 ± 0.22	
8–10	1.16 ± 0.09	1.43 ± 0.16	
10–12	1.30 ± 0.07	1.11 ± 0.09	
12–14	1.26 ± 0.03	1.12 ± 0.20	
14–16	1.36 ± 0.05	1.23 ± 0.12	
16–18	1.28 ± 0.16	1.17 ± 0.04	
18–20	1.28 ± 0.06	1.18 ± 0.12	
20–25	1.46 ± 0.02	1.36 ± 0.08	
25–30	1.16 ± 0.01	1.68 ± 0.05	

137Cs profiles

Detectable 137Cs was measured from the surface to a depth of at least 20–30 cm in all soil profiles (Fig. 3). Total 137Cs inventories of the soil profiles are less than or equal to that expected if the assumed initial activity of 137Cs (0.75 Bq cm−3) remained in place and decayed for a period of 45 years from deposition in 1963 to sampling in 2008. Maximum measured activity for 137Cs was 27.3 ± 0.6 mBq cm−3. Profiles of 137Cs activity generally increase with depth from activities of about 2–6 mBq cm−3 at the surface to maximum activities at around 8–14 cm depth, followed by a general decrease to values of 0.5–2 mBq cm−3 at 30 cm depth.

Figure 3 Depth vs. 137Cs activity.

Depth (cm) vs. average 137Cs activity (Bq cm−3) in cores collected from the ambient (open circles) and elevated-CO2 (filled circles) plots at the Oak Ridge FACE site. Solid lines (gray, ambient; R2 = 0.84; black, elevated-CO2; R2 = 0.75) are best-fit advection-diffusion model profiles based on Eq. (2).

Pre-treatment core samples for depth ranges of 0–5, 0–15, 15–30, and 30–45 cm collected in 1997 from both the pre-treatment ambient and elevated-CO2 plots (prior to initiation of the FACE experiment) had cumulative 137Cs activities equal to those of the post-treatment samples collected in 2008 (Fig. 4). There was no measurable activity of 137Cs beyond 30 cm depth before and during the FACE experiment (Figs. 3 and 4).

Figure 4 Cumulative 137Cs activity.

Average cumulative 137Cs activity (Bq/cm2) vs. depth (cm) in soil cores from ambient and elevated-CO2 plots collected in 2008 (after 10 years of CO2 release) and for two single sets of samples collected from the same locations in 1997 (before the beginning of CO2 release) at the Oak Ridge FACE site.

For the 2008 samples, more 137Cs activity was found at greater depth in elevated CO2 plots when compared to ambient control plots (Fig. 3; p < 0.05). There was measureable 137Cs activity at the 25–30 cm in soil samples collected in the elevated CO2 plots, whereas the average depth of the deepest measurable 137Cs activity in the ambient plots was 20.6 ± 2.5 cm. Similar values of 137Cs activity were found for the 15–30 cm soil samples collected in 1997.

Bioturbation model

Bioturbation derived mixing rates were not significantly different between the ambient and elevated CO2 plots (Table 2; Fig. 3). We solved Eq. (3) for the biogenic diffusivity (Db), particle advection velocity (ν), and feeding rate constant (γ) values. Advection velocities are indistinguishable for both treatments with an average value of 0.18–0.19 cm yr−1 (Table 2). The feeding rate constants were 0.008 ± 0.003 yr−1 for ambient and 0.005 ± 0.001 yr−1 for elevated-CO2 plots. The Db values were at 0.53 ± 0.20 cm2 yr−1 at ambient and 0.63 ± 0.29 cm2 yr−1 at elevated-CO2. These biogenic diffusion coefficients (Db) were used for calculating mixing time constants (τ) for a soil layer thickness L = 20 cm (τ=L2Db−1) (Kaste, Heimsath & Bostick, 2007). The top 20 cm layer of soil at the ORNL FACE site has estimated average mixing times ranging from about 640 to 750 years (Table 2) but with wide spatial variability.

210Pb profiles

The activity ratio (210Pb/226Ra) is a good indicator of excess (or deficient) 210Pb relative to that expected from secular equilibrium with 226Ra (at secular equilibrium, (210Pb/226Ra) = 1). The average (210Pb/226Ra) activity ratio profiles in the ambient and elevated-CO2 plots are similar, showing excess 210Pb in the upper 5-to-10 cm and a deficit of 210Pb below 10 cm depth (Fig. 5). Best-fit solutions of Eq. (4) to the (210Pb/226Ra) profiles all yielded lower values of diffusion constant (near 0) and advection rate (∼0.9 cm yr−1) than did the 137Cs models. We show the best-fit steady-state advection-decay model in comparison with the mean (210Pb/226Ra) profiles in Fig. 5. The parameters in this model were a constant initial (210Pb/226Ra) value of 2.3 and a steady-state (210Pb/226Ra) ratio of 0.75 at depths below 20-to-24 cm, where excess 210Pb has decayed to <2% of its initial amount (Fig. 5). The steady-state value of 0.75, representing a 25% loss of in situ 222Rn production, is based on a survey of 222Rn loss in 119 soil cores from undisturbed landscapes in North America. As with the 137Cs profiles, no significant difference in mean (210Pb/226Ra) profiles is evident between the ambient and elevated CO2 plots.

Discussion

The rate of bioturbation, as indicated by the best-fit biodiffusion coefficient Db from the 137Cs model (Eq. (3)), was indistinguishable between ambient and elevated CO2 conditions (Table 2) despite increased root growth and enhanced earthworm density at the treatment sites (Iversen, Ledford & Norby, 2008; Sánchez-de León et al., 2014). This mixing rate was sufficient to move some SOC from the surface to depth and vice-versa during the 10-year FACE experimental period, suggesting that not all the 13C depleted C seen at a given depth at elevated CO2 is solely derived from C inputs at that depth. This downward movement of FACE-labeled C by bioturbation may partly contribute to the inability to detect relative increases in SOC below 5 cm at elevated CO2 conditions when compared to ambient (Jastrow et al., 2005). Radionuclide profiles of 40K and 226Ra, however, do not show evidence of more rapid leaching of cations at elevated CO2 conditions when compared to the ambient ones, at least in the top 30 cm of soil (Fig. 6). The 40K and 226Ra profiles rather may indicate decomposition of labile SOC in the upper 5 cm of soil, with corresponding enrichment of 40K and 226Ra in the residual, more refractory organic matter (Kaste et al., 2011).

The 137Cs and unsupported 210Pb profiles of this forest resemble those observed in other studies of these radionuclides in soils (Dörr & Münnich, 1989; Dörr & Münnich, 1991; Kaste et al., 2011; Matisoff & Whiting, 2012). Elevated atmospheric CO2 results in soils having substantially higher root biomass (Matamala et al., 2003; Lynch et al., 2013), soil CO2 concentrations and flux (Taneva et al., 2006; Duval et al., 2013), and in some instances increased microbial and soil macrofaunal activity, including that of earthworms (Sánchez-de León et al., 2014; Sánchez-de León et al., 2018). All these factors could enhance the vertical movement of 137Cs and unsupported 210Pb within the soil profile under elevated CO2 conditions. However, the higher microbial activity in the organic rich soil layers (<15 cm for these soils) often seen in response to elevated CO2 conditions ( Gonzalez-Meler & Taneva, 2011; Cheng et al., 2014), could also increase the retention of 137Cs and unsupported 210Pb in the top layer of the soil (Bruckmann & Wolters, 1994). Indirect evidence supports the notion of potentially higher bioturbation in a higher CO2-world. For instance, the feeding constant rate (Table 2) is additive with the 137Cs decay constant in Eq. (3) and thus may indicate some net removal of 137Cs from the profile by leaching or by faunal or root uptake. These biological transport processes also have consequences for carbon burial at centurial time scales that need to be considered in models of the C cycle.

Table 2 Bioturbation.

Parameter values obtained from advection-diffusion model. Parameters Db, v, and γ are derived from best-fits of average 137Cs activity profiles to Eq. (2). The soil mixing time τ is calculated for L = 20 cm. Values are averages of two ambient CO2 and two elevated CO2 rings ± standard deviations.

Symbol	Parameter	Unit	Ambient CO2	Elevated CO2	
Db	Bio-diffusion coefficient	cm2 yr−1	0.53 ± 0.20	0.63 ± 0.29	
v	Advection term	cm yr−1	0.19 ± 0.02	0.18 ± 0.03	
γ	Feeding rate constant	yr−1	0.008 ± 0.003	0.005 ± 0.001	
τ	Soil mixing time (τ = L2Db−1)	yr	750 ± 210	640 ± 200	

Figure 5 Activity ratio (210Pb/226Ra) vs. depth (cm) in soil cores collected in 2008 from ambient (open circles) and elevated-CO2 (filled circles) plots at the Oak Ridge FACE site.

Solid curve (blue) represents constant addition of 210Pb to the surface, an advection rate of 0.185 cm yr−1 based on best-fit of advection-diffusion model (Eq. (3)) to mean 137CS profiles, and decay of 210Pb to a steady-state value of 0.75 × (226Ra), representing 25% loss of in situ 222Rn production. Dot-dashed vertical line represents the typical mean value of 0.75 for soil (210Pb/226Ra) (Graustein & Turekian, 1990). Dashed curve (red) represents constant addition of 210Pb to the surface and an advection rate of 0.0.093 cm yr−1 based on the best-fit of advection-diffusion model (Eq. (4)) to mean unsupported-210Pb profiles, and decay of 210Pb to a steady-state value of 0.75 × (226Ra). Apparent deficiency of excess 210Pb in the soil profiles is consistent with diffusive escape of 222Rn produced in situ, possibly enhanced by bioturbation and transpiration occurring in the shallow root zone.

Figure 6 (A) Depth (cm) vs. 40K activity (Bq cm−1) and (B) depth (cm) vs. 226Ra activity (Bq cm−1) in average soil profiles from the ambient (open circles) and elevated-CO2 (filled circles) plots at the Oak Ridge FACE site.

The net long-term rate at which soil material is moved downward by burial and advective transport is given by the model parameter ν (from 137Cs models this is 0.18 cm yr−1, Table 2, but a value of only about 12 that is indicated by the average unsupported 210Pb profiles). Other studies have shown that 137Cs transport is faster and somewhat decoupled from that of 210Pb (Dörr & Münnich, 1989; Dörr & Münnich, 1991). Over the 10-year duration of the FACE experiment from its initiation in 1998 through our sample collection in 2008, material deposited at the surface (where bulk density is the smallest, Table 1) could be transported by advection to a mean net depth of 1.8 cm. Litter deposited at the surface in the elevated-CO2 profiles should be clearly distinct in terms of its δ13C value, because the CO2 released during the FACE experiment had a much lower δ13C value than that of atmospheric CO2 (Fig. 2). In fact, the top 2 cm of the elevated-CO2 profile clearly has significantly lower δ13C values than the ambient profile (Fig. 1B), and much higher SOC content as well (Fig. 1A). These differences are attributable to the influence of the elevated CO2 treatment during the FACE experiment. The data shown in Fig. 2 indicate, however, that the influence of the 13C-depleted CO2 released to the atmosphere at the FACE site appears to have affected the amount and isotopic composition of bulk SOC throughout essentially the entire 30-cm depth of the elevated-CO2 soil profile. This implies that other processes must have acted to increase inputs and transport of some fraction of SOC downward at a rate higher than that given by the mean net advective transport term in the bioturbation model. Such processes may include bioturbation caused by higher root growth and turnover as well as the feeding activity of burrowing organisms, and advective transport of dissolved inorganic carbon (DIC), dissolved organic carbon (DOC), and particulate organic carbon (POC) in soil pore water, all of which can accelerate the movement of SOC along specific pathways.

Is there an enhancement of weathering activity at elevated CO2? The difference in cumulative 137Cs activity between the ambient and elevated CO2 plots was apparently present before the CO2 experiment started, because it is also seen in the pretreatment soil samples (Fig. 3). Unfortunately, the pretreatment soil samples only indicate total 137Cs to a 30 cm depth but not its distribution along the entire soil profile with the deepest sample being 15–30 cm deep. Based on the 137Cs profile alone, it cannot be ruled out that the CO2 treatment had an effect on the maximum depth of 137Cs activity. The importance of earthworms in bioturbation, and for increasing soil porosity and permeability, has been noted in a number of other studies involving the interpretation of 137Cs profiles of soils (Bunzl, 2002b; Jarvis et al., 2010; Müller-Lemans & Van Dorp, 1996; VandenBygaart et al., 1998). The increases in root, soil flux and earthworm activity seen in the elevated CO2 treatment at this site are consistent with this potential effect of CO2 on maximum depth of 137Cs. Transport of 137Cs could be enhanced by increases in porosity and permeability of soil caused by earthworm activity, which could increase the rate and volume of fluid flow through the soil.

Another mechanism for increasing downward transport of both 137Cs and C could be the acidification of soils because of enhanced soil metabolic activity often seen at elevated CO2 (e.g.,Taneva et al., 2006; Hopkins et al., 2013). Increases in soil partial pressure of CO2 (pCO2) in plots exposed to elevated CO2 may decrease soil pH and alter metal chemistry (Natali et al., 2008). Lower pH caused by higher pCO2 can increase soil weathering rate along with desorption of adsorbed cations, production of bicarbonate by carbonic acid neutralization, and consequently enhanced advective transport of desorbed cations and DIC deeper into the soil profile (Andrews & Schlesinger, 2001). The depth profiles of endogenous soil 40K and 226Ra species show nearly constant concentration of each nuclide with depth in both the ambient and elevated-CO2 plots (Fig. 6). This may indicate that the change in soil pH associated with elevated CO2 was not sufficient to cause a substantial increase in weathering rate and mobilization of K+ and Ra2+ ions within the top 30 cm of the soil profile. These two ions could be mostly incorporated within mineral grains, whereas 137Cs is associated with mineral surfaces and is therefore more susceptible to desorption (Dörr & Münnich, 1989; Dörr & Münnich, 1991). However, the results presented here do not provide evidence of increased weathering rates in the top 30 cm in response to elevated CO2 as suggested elsewhere (Cheng et al., 2010).

Soil mixing dynamics

There is a rapid decrease in (210Pb/226Ra) ratios in the ambient and elevated-CO2 plots at the FACE site, relative to that predicted by the simple 210Pb advection-decay model (Fig. 5). The constant deposition of 210Pb from the atmosphere to the soil surface creates a condition of radioactive disequilibrium where 210Pb in the shallow parts of soil profiles is in excess of that produced in situ by decay of 226Ra and intermediate daughters. The profile of excess 210Pb can be modeled in terms of soil or sediment accumulation and erosion rates and mixing parameters (Olsen et al., 1981; Robbins, 1986; Kaste, Heimsath & Bostick, 2007; Kaste et al., 2011; Matisoff & Whiting, 2012). The 210Pb/226Ra profiles depicted in Fig. 5 decay too rapidly with depth to be consistent with simple downward advection at 0.18 cm yr−1 and radioactive decay of 210Pb. Using Eq. (4), we obtained a best-fit value for advective transport (v) of about 0.9 cm yr−1. The relatively low and constant value of the 210Pb/226Ra activity ratio at depth indicates diffusive escape of 222Rn (likely via transpiration stream and soil porosity) that was produced in situ from 226Ra decay. This 222Rn escape is possibly enhanced by bioturbation and transpiration occurring in the shallow root zone (top 10 cm) where the bulk density is the lowest (Table 1). The 137Cs profiles and the rapid decrease in excess 210Pb suggest a distinct soil boundary at about 4 cm deep, below which most of the 137Cs activity resides (Fig. 5). This 0–4 cm depth soil layer is also evident in isotope profiles shown in Figs. 1 and 6. This soil multi-isotope boundary at 4 cm depth is consistent with the enhanced SOC accumulation at elevated CO2 conditions when compared to ambient seen at the site between 0 and 5 cm (Fig. 1 of Jastrow et al., 2005). Further, these results are also consistent with the presence at the site of endogeic earthworms (Sánchez-de León et al., 2018), which avoid the soil surface likely preventing predation or competition with litter layer fauna.

A second soil isotope boundary is detected at about 16 cm deep, where the 210Pb/226Ra activity ratio approaches the typical disequilibrium value of 0.75 seen in deeper soils (Graustein & Turekian, 1990) (Fig. 5). This 4–16 cm soil section may represent a soil mixing compartment influenced by root proliferation and earthworm activity, potentially redistributing and homogenizing SOC concentration within this depth range (Fig. 1A). This may partly prevent detection of soil C accrual in response to elevated CO2 levels at these depths (Fig. 1A), despite the isotopic evidence for input of new SOC (Fig. 1B). More research using multiple radioisotope tracers to detect soil profile mixing sections may allow better determinations of C dynamics than are possible by using the traditional arbitrary depth comparisons.

Conclusion

The 137Cs profile and the associated bioturbation models represent a 45-year period between the deposition of the 137Cs bomb-spike in 1963 and the collection of the soil cores in 2008. The parameter values obtained from the advection-diffusion models are not significantly different between the average ambient and elevated-CO2 profiles (Table 2). Elapsed time from the beginning of the FACE experiment to the time of sampling was inadequate to cause a substantial response in terms of observable soil bioturbation. Future studies require greater sample density to acquire sufficient statistical evidence for observing subtle changes in such heterogeneous systems. However, even during the decade-long duration of the FACE experiment there is significant evidence for increased inputs of new SOC at soil depth and for enhanced migration of SOC beyond 20 cm depth, likely caused by both biological and chemical processes related to elevated CO2. The multiple isotopic tracer approach used here indicated at least two soil compartments: one in the top 4 cm where C accumulated in response to CO2 and one below 4 cm where SOC may have been redistributed. These “soil mixing cells” may bring into question the traditional depth comparisons for SOC (e.g., 0–5 cm, 0–10 cm) that are routinely done in soil studies, and which may obscure detection of SOC changes in response to environmental factors.

The biological and geochemical effects of elevated atmospheric CO2 could have substantial consequences for carbon burial and the fate of deep soil C over a longer time scale (centuries) than usually considered in soil C and Earth System models (Schmidt et al., 2011; Kaste et al., 2011; Todd-Brown et al., 2013). For instance, in a typical 100–200-year model run, C residing in a given soil layer could move downward by bioturbation and weathering to another soil layer depicted in a model (e.g., CLM4.5), affecting rates of decomposition, soil C turnover times and the soil feedback on the atmospheric concentration of CO2. Bioturbation processes could substantially increase the net long-term storage of soil C and should be incorporated in soil-atmosphere interaction models.

Supplemental Information

Supplemental Information 1 Raw data for radioisotopes and soil bulk density

Raw data for 137Cs, 210Pb, 40K, 226Rn, and soil bulk density under elevated CO2 concentration.

Click here for additional data file.

Supplemental Information 2 Raw data for carbon percentage (%) and carbon isotope ratio (d13C/d12C)

Raw data for concentration of carbon (%) in soil and carbon isotope ratio (d13C/d12C) in soil under elevated CO2 concentration.

Click here for additional data file.

Dr. Sánchez-de León thanks the Department of Biological Sciences at UIC for support. We thank Javier Lugo-Perez, Jessica Rucks and Elena Blanc-Betes for assistance during soil processing and isotope analyses and David H. Wise for useful comments.

Additional Information and Declarations

Competing Interests

Author Contributions

Data Availability

Miquel Angel Gonzalez-Meler is an Academic Editor for PeerJ.

Miquel A. Gonzalez-Meler and Neil C. Sturchio conceived and designed the experiments, performed the experiments, analyzed the data, contributed reagents/materials/analysis tools, prepared figures and/or tables, authored or reviewed drafts of the paper, approved the final draft.

Armen Poghosyan performed the experiments, approved the final draft.

Yaniria Sanchez-de Leon conceived and designed the experiments, contributed reagents/materials/analysis tools, approved the final draft.

Eduardo Dias de Olivera analyzed the data, authored or reviewed drafts of the paper, approved the final draft.

Richard J. Norby contributed reagents/materials/analysis tools, authored or reviewed drafts of the paper, approved the final draft.

The following information was supplied regarding data availability:

The raw data are provided in the Supplemental Files.

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
