# Peer review of "Does elevated atmospheric CO2affect soil carbon burial and soil weathering in a forest ecosystem?"

_PeerJ, doi:10.7717/peerj.5356_

## Round 0.1 · original submission · Minor Revisions

In particular, Rev#2 poses the question that the diffusion term needs to be also considered in a quantitative manner and, consequently, that the model has to be slightly modified in order to include it.

Reviewer 1 ·

Basic reporting

English could be improved in places, literature references and explanation of previous work currently incomplete.

Experimental design

I believe that the experimental design is sound and the research question is well defined.

Validity of the findings

Generally sound-I disagree with the 100% radon loss model but their interpretations are reasonable. . see comments below

Additional comments

Review of “Does forest growth at elevated CO2 affect bioturbation and soil carbon burial rate?” by Gonzalez-Meler et al.
In this manuscript, the authors show stable carbon isotope and radionuclide tracers of organic carbon transport in soil profiles measured at the Free Atmosphere Carbon Exchange (FACE) site which allows them to test the effect of high CO2 and ambient CO2 conditions on burial and bioturbation. Overall I find the study to be interesting and novel and the dataset is sound. The authors tackle a timely and significant problem: how rising atmospheric CO2 concentrations might affect soil processes, and possibly drive a feedback involving carbon burial. The paper is written reasonably well and is easy to follow but there are some places where the English could be tightened. Some of the background and context also need improvement. Finally, the possibility that 137Cs transport is at least partially decoupled from organic matter transport should be more adequately addressed.
Below I give more specific comments/suggestions:
1) Abstract: the first part of the first sentence is a bit awkward: “studies of effects…” should read “studies measuring the effects”
2) I don’t believe that “bioturbation” needs to be defined in the abstract- this is a standard term
3) Introduction line 50: the words “accessibility of” should be removed, and so the statement should simply be “Soil organic carbon decomposition depends on….”
4) The in-text referencing is inconsistent: sometimes the authors list multiple authors, and other times they use “et al”. See lines 63, 73, and 94 for examples that need to be fixed.
5) More background on previous research using fallout radionuclides as tracers of organic matter transport in soils are needed. The authors reference some studies but don’t really explain the work that has been established. For example, Dӧrr and Munnich (1989, 1991) were I believe the first scientists to use radionuclides (14C, 210Pb, 137Cs) to trace soil organic matter transport in soils yet they are not referenced here. They found that 210Pb was a good tracer of soil carbon transport, but that Cs was too soluble and seemed to have an additional transport mode that was probably dissolved Cs+ cation. I believe that their results may be applicable to the current study. Kaste et al. (2011) showed that 210Pb was a good tracer of soil organic matter burial and organic matter age in Podzols collected from both Norway and the northeastern United States. I find the introduction and context of the manuscript to be somewhat incomplete with regards to describing previous work.
6) Material and Methods: could be improved. Describe the coring device a bit more- was it a plastic tube?
7) The authors fit two different models to their 210Pb/226Ra data- one that assumes 25% of 222Rn loss from soils, and the other that assumes 100% loss of 222Rn from soils. While I agree that the loss of 222Rn is probably depth-dependent, and that bioturbation/weathering might increase the loss at the surface, I do not think that 100% 222Rn loss is realistic. Most studies show that the loss is closer to 10-25% (Graustein & Turekian). While assuming 100% loss of 222Rn at the surface permits a stronger fit to the data in the upper 10cm, a more likely scenario is that the diffusion and advection rates are variable with depth. I suggest that the authors address this possibility more thoroughly in the discussion.
8) The reference list at the end of the manuscript needs to be checked carefully for formatting.
Ultimately I believe that this is a very novel and useful study. I hope to see it published after some revisions.

Suggested References:

Using atmospheric fallout to date organic horizon layers and quantify metal dynamics during decomposition
By: Kaste, James M.; Bostick, Benjamin C.; Heimsath, Arjun M.; Steinnes E., and Friedland AJ
GEOCHIMICA ET COSMOCHIMICA ACTA Volume: 75 Issue: 6 Pages: 1642-1661 Published: MAR 15 2011

LEAD AND CESIUM TRANSPORT IN EUROPEAN FOREST SOILS
By: DORR, H; MUNNICH, KO
WATER AIR AND SOIL POLLUTION Volume: 57-8 Pages: 809-818 Published: AUG 1991

DOWNWARD MOVEMENT OF SOIL ORGANIC-MATTER AND ITS INFLUENCE ON TRACE-ELEMENT TRANSPORT (PB-210, CS-137) IN THE SOIL
By: DORR, H; MUNNICH, KO
Conference: 13TH INTERNATIONAL RADIOCARBON CONF Location: DUBROVNIK, YUGOSLAVIA Date: JUN 20-25, 1988
RADIOCARBON Volume: 31 Issue: 3 Pages: 655-663 Published: 1989

Reviewer 2 ·

Basic reporting

This article is well written and the English is of a high quality. For the most part it is clear and easy to follow. I have a few minor comments on clarity:
Line 53-55. I find this sentence rather long and difficult to process. I would suggest to cut it down to:
"Climate change forcing factors can directly and indirectly affect soil C stocks, altering the resilience of vegetation and human society to climate change."
Lines 75. Saying that previous studies do not link long term soil C movement to 'climate change forcing factors' implies that that will be done so here. To me that implies the impact of climate change, and I do not consider a CO2 change as a change in climate (in fact, a lot of model studies 'climate' and 'CO2' are considered as two different forcings). Perhaps you could say something like 'climate change or CO2 change' instead of 'climate change forcing factors', it might just be clearer. Hope that makes sense.

Figure 4 is missing and has been replaced with a copy of Figure 1. Please address this. So I cannot comment on Figure 4 but otherwise the figures appear to be appropriately presented.

The hypothesis is that elevated CO2 could enhance bioturbation and/or soil weathering, and the analysis addresses this, thus the study is nicely self contained. My one concern is that the title does not very clearly reflect the hypothesis. The study considers more than one potential impact of elevated CO2, including direct effects on weathering and therefore not only the impact of 'forest growth'. I would suggest removing 'forest growth' and perhaps something along the lines of:
'Does elevated CO2 affect soil carbon burial and soil weathering in a forest ecosystem?' (it wouldn't have to be exactly this!)

Experimental design

In general the work seems to be original, well-defined and relevant. I cannot comment too much on the methods since I don't have a large experience with experimental methods.

Method description:
Line 174. Perhaps this is commonly known but I had not heard of a 'feeding rate constant' so I would like to see a few words here to explain what this represents.

Regarding the experimental design, since the differences in some of the key variables were not statistically significant and yet could have been reasonably large (for example, the mean mixing rate, which directly addresses the hypothesis, was 0.63 cm^2/yr in elevated CO2 and 0.53 cm^2/yr in ambient CO2, but with error bars too large to distinguish them), taking a larger number of soil samples might allow the potential differences (or lack of difference) to be resolved. I would suggest adding some comments in the discussion on this and potentially a call for additional sampling in further studies.

Validity of the findings

My main concern with this analysis is that after fitting a model with both advection and diffusion terms, only the advection term is then considered. For example on line 288-290, the advection rate is considered but not the diffusion rate. With a diffusion rate of 0.5 cm2/year, that could certainly result in 13C-depleted C reaching a greater depth than from the advection term alone. The advection process relates more to soil accumulation and burial, whereas the diffusion term will relate to the bioturbation (e.g. worms) which will mix some of the surface carbon to depth (and deep C to the surface). Even considering both terms, it is still likely that the other processes mentioned on the following lines can also have an impact, but definitely the diffusion term needs to be also considered, and in a quantitative manner (i.e. how much of the deeper C might this account for). Then the abstract lines 37-39 will also need some modification.

Another concern is the conclusion that mixing may prevent detection of the additional carbon in the layer (for example 347-349). While this could be a factor, could it also be that both control and elevated CO2 plots have carbon addition at those depths, but in the elevated CO2 plot it is depleted C? Surely the presence of depleted C does not on its own imply that the C addition has been greater than in the control?

Finally just a question regarding the Pb/Ra profile shown on Figure 5 - the conclusion is that it decays too rapidly with depth to be consistent with the advection term implied from the 137Cs profile, and therefore there must be some additional losses. However, 137Cs is more prone to desorption than 210Pb, and if the 137Cs is to some extent desorbed and leached downwards, could this also be providing additional advection which would not be seen for the 210Pb?

Line 367-368, "Our results indicate..." I think this sentence is too strong a claim given that the results did not show much of an impact.
Line 259 "do not show unequivocal evidence" I would change this to "do not show evidence", because not only is it not unequivocal, it simply doesn't appear to be there.

Additional comments

Line 201. I think this should say g/cm^3 and not kg/cm^3!
Line 259. Says 222Ra, should be 226Ra?

---

## Round 0.2 · accepted · Accept

I confirm that you have fulfilled the changes requested by the reviewers and your manuscript is now suitable for publication in PeerJ.